# Rapid Antibacterial Activity of Cannabichromenic Acid against Methicillin-Resistant *Staphylococcus aureus*

**DOI:** 10.3390/antibiotics9080523

**Published:** 2020-08-16

**Authors:** Maria Galletta, Tristan A. Reekie, Gayathri Nagalingam, Amy L. Bottomley, Elizabeth J. Harry, Michael Kassiou, James A. Triccas

**Affiliations:** 1Discipline of Infectious Diseases and Immunology, School of Medical Sciences, University of Sydney, Sydney, NSW 2006, Australia; mgal9502@uni.sydney.edu.au (M.G.); gayathri.nagalingam@sydney.edu.au (G.N.); 2Charles Perkins Centre and Marie Bashir Institute for Infectious Diseases and Biosecurity, University of Sydney, Sydney, NSW 2006, Australia; 3School of Chemistry, University of Sydney, Sydney, NSW 2006, Australia; Tristan.Reekie@anu.edu.au; 4The Ithree Institute, University of Technology Sydney, Ultimo, NSW 2007, Australia; amy.bottomley@uts.edu.au (A.L.B.); Elizabeth.Harry@uts.edu.au (E.J.H.)

**Keywords:** methicillin-resistant *Staphylococcus aureus*, antibiotics, phytocannabinoids, cannabichromenic acid, time-kill assay

## Abstract

Methicillin-resistant *Staphylococcus aureus* (MRSA) has proven to be an imminent threat to public health, intensifying the need for novel therapeutics. Previous evidence suggests that cannabinoids harbour potent antibacterial activity. In this study, a group of previously inaccessible phytocannabinoids and synthetic analogues were examined for potential antibacterial activity. The minimum inhibitory concentrations and dynamics of bacterial inhibition, determined through resazurin reduction and time-kill assays, revealed the potent antibacterial activity of the phytocannabinoids against gram-positive antibiotic-resistant bacterial species, including MRSA. One phytocannabinoid, cannabichromenic acid (CBCA), demonstrated faster and more potent bactericidal activity than vancomycin, the currently recommended antibiotic for the treatment of MRSA infections. Such bactericidal activity was sustained against low-and high-dose inoculums as well as exponential- and stationary-phase MRSA cells. Further, mammalian cell viability was maintained in the presence of CBCA. Finally, microscopic evaluation suggests that CBCA may function through the degradation of the bacterial lipid membrane and alteration of the bacterial nucleoid. The results of the current study provide encouraging evidence that cannabinoids may serve as a previously unrecognised resource for the generation of novel antibiotics active against MRSA.

## 1. Introduction

Microbial drug resistance has proven to be one of the greatest and most imminent threats to public health. Over the last half century, antibiotic-resistant bacteria have evolved at an alarming rate, reaching epidemic proportions globally. Of particular concern is the rise and spread of methicillin-resistant strains of *Staphylococcus aureus* (MRSA). MRSA is the most commonly identified antibiotic-resistant pathogen in many parts of the world, with 50% of all nosocomial infections proposed to result from MRSA [1]. Such MRSA infections incur the largest annual cost to the U.S. health system of any acute infectious disease, yielding an economic burden of approximately $USD 2.2 billion annually [2]. Further, more than 19,000 deaths in the U.S. in 2017 were due to invasive MRSA infections [3]. Finally, MRSA has also been associated with life-threatening infections typically uncommon of *S. aureus*, including necrotising fasciitis [4] and necrotising pneumonia [5].

The success of MRSA can be attributed to its propensity for the development of antibacterial resistance. MRSA has demonstrable resistance not only to methicillin but to the entire class of β-lactam antibiotics [6]. Further, MRSA isolates concomitantly display resistance to the aminoglycosides, macrolides, fluoroquinolones, and the chloramphenicols [7]. Long-term therapies with antibiotics of alternate classes such as the lincomycins or tetracyclines are often implemented to treat localised MRSA infections [8]. Vancomycin, of the glycopeptide class of antibiotics, remains the cornerstone of treatment for systemic MRSA infections; nevertheless, its use is limited to invasive infections to minimise the generation of resistance and as it is associated with suboptimal outcomes, principally due to the cytotoxic complications of prolonged and intense treatment regimens [8]. However, despite such efforts, isolates with intermediate and complete resistance to vancomycin are well described [9].

Whilst novel and more recently approved antibiotics for the treatment of MRSA infections have been identified [10], only a handful have proven to be both successful and widely applicable in the clinical setting. These include linezolid, of the oxazolidinones class of antibiotics [11], and daptomycin, a cyclic lipopeptide [12]. Resistance rates to both linezolid and daptomycin continue to remain low [13]. Nevertheless, dependence upon these limited range of antibiotics will systematically increase selective pressures and encourage the evolution of further antibiotic-resistant mutants. Consequently, the quest for therapeutics effective against this aggressive human pathogen remains vital. 

The cannabinoids are a class of molecules responsible for the modulation of emotion, appetite, memory, and immunity through the stimulation of in vivo cannabinoid receptors [14]. Cannabinoids are classified based on their origins. The natural cannabinoids refer to the endocannabinoids, produced endogenously within humans, and the phytocannabinoids, produced by the *Cannabis sativa* plant [14,15]. The synthetic cannabinoids refer to synthetically produced compounds that demonstrate activity at the cannabinoid receptors1 [15]. The phytocannabinoids have been identified for their mood-enhancing and psychoactive effects [16]. Medicinally, however, these compounds are recognised for their analgesic activity and thus medicinal applications in various neurological and musculoskeletal disorders [14]. Previous work has revealed that cannabinoids possess inhibitory activity against a range of bacteria and fungi [17], suggesting an antimicrobial potential for these compounds. Indeed, recent efforts have described the antibiotic potential of phytocannabinoids, with a particular focus on component cannabigerol (CBG) [18]. In this manuscript, we characterise the ability of the phytocannabinoid cannabichromenic acid (CBCA) and its related synthetic analogues to successfully inhibit the growth of MRSA and other clinically relevant pathogenic bacteria. 

## 2. Results and Discussion

### 2.1. Phytocannabinoid Analogues Inhibit the Growth of Drug-Resistant and Pathogenic Bacteria

Recent synthetic efforts have permitted access to previously unavailable phytocannabinoids, including CBCA and a group of chemically related analogues [19,20]. Following such access, this group was analysed for antibacterial activity against a selected range of clinically relevant gram-positive bacteria (Table 1). The determination of MIC revealed the potent antibacterial activity of CBCA against MRSA, methicillin-sensitive *S. aureus* (MSSA), and vancomycin-resistant *Enterococcus faecalis* (VRE) (Table 1). CBDVM revealed activity against MRSA (Table 1). 

To determine any cytotoxic effect of CBCA and CBDVM against the mammalian A549 (human alveolar basal epithelial cells) and HepG2 (human liver cancer) cell lines, the minimum toxicity concentration (MTC), defined as the minimum amount of compound needed to inhibit growth of mammalian cells, was calculated. For both cell lines, CBCA demonstrated limited toxicity (>100 μM), with MTC values considerably higher than this compound’s MIC value against MRSA (Table 1). CBDVM revealed similar patterns of toxicity as CBCA against the mammalian cells.

### 2.2. CBCA Exerts Rapid Bactericidal Activity that Is Independent of Bacterial Cell Density and Metabolism

Due to its potent antimicrobial activity in the absence of significant toxicity against mammalian cells, CBCA was selected for further study. A time-kill assay was employed to determine the kinetics of bacterial inhibition afforded by this compound. Vehicle-treated cultures grew rapidly, with a 3-log increase in the CFU/mL of viable bacteria over the 24-h period (Figure 1A). Vancomycin exhibited potent and prolonged bactericidal activity, demonstrating a >2-log reduction in the CFU/mL of viable MRSA following 8 h of treatment and undetectable levels of bacteria at 24 h (Figure 1A). CBCA demonstrated relatively more rapid bactericidal activity than vancomycin, reducing the number of viable bacteria to undetectable levels at 2 h post treatment (Figure 1A). At 24 h, however, the concentration of viable bacteria treated with CBCA demonstrated a >5-log increase.

Whilst CBCA revealed potent antibacterial activity, in in-vivo infections bacteria often reach considerably larger cell densities (as great as ~10^10^ cells per infected individual) [21]. Further, antibiotics typically fail to exert any significant antibacterial effects against an infection of such magnitude [22]. Thus, it was next determined whether CBCA would maintain its potent bactericidal activity against a significantly higher bacterial load (600-fold greater than starting inoculum). Under these conditions, vancomycin exhibited poor bactericidal activity, yielding a 1-log reduction in the number of viable bacteria over the 24 h period (Figure 1B). CBCA elicited a >4-log reduction in the bacterial load from 2 to 8 h of treatment (Figure 1B), indicating rapid bactericidal activity that is independent of cell density. At 24 h, the number of viable bacteria in this culture rose by approximately 4-log CFU/mL. The consistent recovery in the number of viable cells at 24 h in CBCA-treated cultures (Figure 1) was hypothesised to indicate a loss of compound activity at extended timepoints or the rapid development of compound resistance. The latter possibility was excluded as bacteria recovered after 24 h demonstrated no change in antimicrobial susceptibility compared to the inoculum (data not shown). 

The linear growth curve of vehicle-treated MRSA cells in high-density cultures suggested that the bacterial cells of this experiment had entered into the stationary phase of growth (Figure 1B). If so, this would suggest the rapid and potent antibacterial activity of CBCA that is independent of cellular metabolism. This suggested potential was further explored using a modified time-kill assay, where bacteria were treated with carbonyl cyanide *m*-chlorophenylhydrazone (CCCP) to artificially arrest cellular metabolism [23]. CCCP addition successfully arrested MRSA growth (Figure 2). Under these conditions, vancomycin was unable to reduce bacterial burden, eliciting less than a 1-log reduction across the 24 h period (Figure 2). In contrast, CBCA reduced bacterial load to undetectable levels at 4 h post treatment, with viable bacteria in this culture remaining undetectable at 24 h post treatment (Figure 2). This observed activity of CBCA against both exponential-and stationary-phase MRSA suggests comprehensive clinical potential for this compound. The stationary phase of growth is vital to MRSA infections, since biofilms, important during nosocomial MRSA infections, are comprised of growth-arrested cells [23,24]. Further, *S. aureus* virulence factor expression is restricted to the stationary phase of growth [25]. Finally, vancomycin and additional antibiotics approved for the treatment of MRSA infections are proven ineffective against growth-arrested cells, justifying both the frequent therapeutic failure of these therapies [26] and our observed inactivity of vancomycin against high-dose MRSA inoculums (Figure 1B). Thus, compounds such as CBCA that can effectively inhibit bacteria in the stationary phase should serve as leading candidates for further pre-clinical assessment.

### 2.3. CBCA Induces Distinct and Degradative Morphological Changes Indicative of Bacterial Degeneration and Cell Lysis

To further define the antibacterial effects of CBCA, phase-contrast and fluorescence microscopy was performed. This microscopic analysis was performed with *Bacillus subtilis*, a rod-shaped bacillus that permits greater observation of morphological changes including alterations in cellular shape and length. CBCA was active against *B. subtilis* in the low micromolar range (MIC = 4 µM). *B. subtilis* cells were incubated with various concentrations of CBCA and imaged at 30 min post treatment. The effects on bacterial growth and cell division were determined through measurement of the lengths of individual cells. A dose-dependent reduction in mean bacterial length was observed following treatment with the highest dose of CBCA (0.4 × MIC; 1.6 µM) (Figure 3). Dose-dependent bacterial cell lysis was also observed in the treated cells (Figure 4), confirming the rapid bactericidal activity observed in the time-kill experiments. Indeed, the movement of insoluble intracellular material into the extracellular environment through apparent pores in the cell wall was imaged (Figure 4B), as were cells in the process of lysis (Figure 4C). Lysed cells, indicated by ‘light grey rods’ or remnants of the cell wall sac, were also observed (Figure 4D). The rapidity and expanse with which CBCA impairs cellular growth and induces lysis is clinically significant as prolonged treatment regimens with antibiotics can, through increased exposure to the drug, promote bacterial resistance [6,27]. Therefore, the fast-and wide-acting nature of CBCA may enable reduced treatment time to help prevent the development of antimicrobial resistance to this compound.

Moreover, despite the loss of intracellular material in the treated bacteria, the rod shape of the *Bacillus* species appeared to be sustained. Interestingly, the phase-contrast images of bactericidal antibiotics active against MRSA and which target peptidoglycan, such as the penicillins, depict a characteristic ‘smear’ of cellular debris. This is due to an inability to withstand osmotic pressure and the subsequent ‘popping’ of the bacteria [28]. The preservation of the rod shape of CBCA-treated *B. subtilis* cells suggested the maintenance of the peptidoglycan wall and, by deduction, lysis through the impairment of the bacterial lipid membrane. This is crucial as the peptidoglycan layer is a common target of antibiotics in use for the treatment of MRSA infections. Targeting of the bacteria lipid membrane would thus provide an alternate mechanism of action with a reduced chance of existing or cross-antimicrobial resistance.

To further investigate the impact of CBCA on cell structure integrity, *B. subtilis* cells were incubated with CBCA and AM466, a lipophilic styryl dye that is incorporated into lipid-containing cellular structures [29]. This was followed by incubation with DAPI to stain nucleic material [30]. Untreated *B. subtilis* cells, as expected, were rod shaped cells with round, defined nucleoids (Figure 5A). Septa were also seen within and between individual cells, confirming the exponential growth of cells. Incubation with CBCA induced the spindling and smearing of nucleic material (Figure 5B) and abrogation of the lipid membrane (Figure 5D). The amalgamation of fluorescence and phase-contrast images proposes maintenance of the peptidoglycan layer simultaneous to cell lysis, the latter suggested by the extracellular localisation of bacterial DNA (Figure 5H). Thus, this finding further suggests that CBCA impaired the structural integrity of the bacterial lipid membrane to result in bacterial cell death (Figure 5H).

Interestingly, antimicrobial peptides, such as the bacteriocins and peptide antibiotics including daptomycin, function by intercalating with the bacterial lipid membrane to form pores and to induce a number of downstream effects which altogether culminate in cell lysis [31,32]. One of these downstream effects includes the binding to bacterial DNA and the inhibition of its replication [33]. Whilst it remains unclear whether CBCA is directly targeting the lipid membrane, treatment with this compound induced rapid degradation of such and led to downstream impacts on nuclear material and cell viability akin to that of a bacteriocin or peptide antibiotic. Further pre-clinical analysis is therefore required to assess this potential mechanism of action.

## 3. Materials and Methods

### 3.1. Bacterial Strains

For the antimicrobial evaluation, three clinical isolates were utilised in this study. The MRSA strain was obtained from the blood culture of a bacteraemic patient at the Royal Prince Alfred (RPA) Hospital, Sydney, Australia. Methicilin-resistance was confirmed following the detection of the mecA genetic material after genetic analysis. The MSSA (34397) strain obtained from a left elbow biopsy of a patient at the RPA hospital, Sydney, Australia. Methicilin-suscpetibility was confirmed following non-detection of the mecA genetic material after genetic analysis. The VRE strain was obtained from a patient at the Westmead Hospital, Sydney, Australia. Vancomycin-resistance was suspected following infection persistence after the administration of adequate vancomycin-therapy. For microscopic analysis, a prototypical Bacillus subtilis (*B. subtilis*) (168) strain was utilised [34].

### 3.2. Cannabinoid Compounds

The compounds cannabichromenic acid [(±)-CBCA], cannabichromene methyl ester trifluoroacetate [(±)-CBCTFA], cannabicyclol methyl eseter [(±)-CBLM], cannabichromene methyl ester [(±)-CBCM] and cannabidivarin methyl ester (CBDVM) were synthesised according to reported procedures [19,20] and were assessed as >95% purity via HPLC analysis.

### 3.3. Bacterial Growth Conditions

Bacterial strains were stored at −80 °C and thawed as needed. All bacterial strains were grown at 37 °C in Luria-Bertani (LB) broth (Difco^TM^ Laboratories, Detroit, MI, USA) with 1% (*w*/*v*) tryptone, 0.5% (*w*/*v*) NaCl and 0.5% yeast extract.

### 3.4. Mammalian Cell Growth Conditions

Mammalian cell lines [human alveolar basal epithelial cells (A549) (ATCC CCL-185) or human liver epithelial cells with hepatocellular carcinoma (HepG2) (ATCC Hb-8065)] were grown at 37 °C and 5% CO_2_ in Dulbecco’s modified eagle media (DMEM) (Lonza, Basel, Switzerland) with 10% (*v*/*v*) heat-inactivated sterile foetal bovine serum (Scientifix Life, North Ryde, Australia), (100 U penicillin + 100 µg/mL) Penstrep (Lonza, Basel, Switzerland).

### 3.5. Resazurin Reduction Assay for Minimum Inhibitory Concentration

All compounds were initially prepared as 25 mM stocks in 100% DMSO and then adjusted to the required concentration in triple distilled water (TDW). The compounds were serially diluted in halving concentrations in TDW and incubated in LB broth overnight at 37 °C with MRSA previously diluted to OD_600_ 0.001. The cells were incubated with 0.05% (*w*/*v*) resazurin sodium salt (Sigma-Aldrich, Castle Hill, Australia) in TDW for 30–60 min. Bacterial growth was quantified by detection of fluorescence at 590 nm using the Infinite M1000 Pro Plate Reader (Tecan, Maanedorf, Switzerland) recording fluorescence at 590 nm. The lowest concentration of compound that resulted in inhibition of bacterial growth was used to determine the minimum inhibitory concentration (MIC).

### 3.6. Resazurin Reduction Assay for Minimum Toxicity Concentration

One ×10^5^ cells were and cultured overnight in DMEM and at 37 °C with 5% CO_2_. The cells were washed twice with phosphate buffered saline (PBS) (Astra Scientific, Gymea, Australia) and fresh media was added. The compounds were serially diluted by halving concentrations in media and the cells were incubated for 4 days. Then, 0.05% resazurin was added to the cells for 4 h and fluorescence recorded at 590 nm using the Infinite M1000 Pro Plate Reader (Tecan, Maanedorf, Switzerland). The lowest concentration of compound that resulted in inhibition of cell growth was used to determine the minimum toxicity concentration (MTC).

### 3.7. Time-Kill Analysis of Compound Inhibition

A bacterial suspension of MRSA in LB broth (OD_600_ of 0.001 or 0.6) was treated with 10 × MIC of CBCA, vancomycin or DMSO alone. In some experiments, the bacteria were treated with cyanide m-chlorophenylhydrazone (CCCP) (10 µM) (SIGMA, Australia) for 1 h to chemically arrest bacterial metabolism. Aliquots were collected from each treatment after 0, 2, 4, 6, 8, and 24 h of incubation at 37 °C with 5% CO_2_, washed twice and resuspended in PBS. Tenfold serially diluted suspensions were plated on LB agar plates and incubated at 37 °C with 5% CO_2_ for 24 h. Individual isolated colonies on the plates were counted and expressed as colony forming units per mL of culture (CFU/mL).

### 3.8. Phase-Contrast and Fluorescence Microscopy

*B. subtilis* cells in exponential phase (OD_600_ 0.1–0.3) were treated with the indicated concentrations of compounds or left untreated (control). At 30 min, aliquots were retrieved, centrifuged (16,000 rcf for 3 min) resuspended in PBS. This bacterial suspension was pipetted onto 2% (*w*/*v*) agarose pads for phase-contrast microscopy using the Zeiss Axioplan 2 Upright Light Microscope.

For fluorescence microscopy, bacterial cells in exponential phase (OD_600_ 0.1–0.3) were treated with indicated concentrations of compounds and incubated for 30 min. For the final 10 min of incubation, AM466 was added to a final concentration of 25 µg/mL. At 30 min, aliquots were retrieved and subjected to the centrifugation and resuspension protocol as described above. DAPI (final concentration of 200 ng/mL) was subsequently added to bacterial suspensions. Following a further 5-min incubation, these suspensions were pipetted onto 2% agarose pads for fluorescence microscopy using the Zeiss Axioplan 2 Upright Light Microscope at 100 × magnification using a Plan ApoChromat (100 × NA 1.4; Zeiss) phase-contrast objective and an AxioCam MRm cooled charge-coupled-device (CCD) camera. All images were analysed using Zeiss AxioVision version 4.8.

## 4. Conclusions

We have identified a previously inaccessible phytocannabinoid compound, CBCA, to be effective against the increasingly prevalent and virulent bacterial pathogen MRSA. CBCA was found to be as efficacious as the current standard of care, the ‘last resort’ antibiotic vancomycin, at inhibiting the growth of MRSA. This activity was proven to be independent of bacterial cell load and metabolism, likely to enhance the clinical utility of this compound. Moreover, the compound’s rapid degradation of the bacterial lipid membrane, with subsequent cell lysis, further proposes promising clinical utility. Additional investigation to elucidate this compound’s mechanism of action, pharmacodynamic properties, and in vivo activity is required. Nevertheless, in an age of a waning antibiotic armamentarium, the discovery of a unique compound with anti-MRSA activity undoubtedly serves as an encouraging and universally welcomed finding.

## Figures and Tables

**Figure 1 antibiotics-09-00523-f001:**
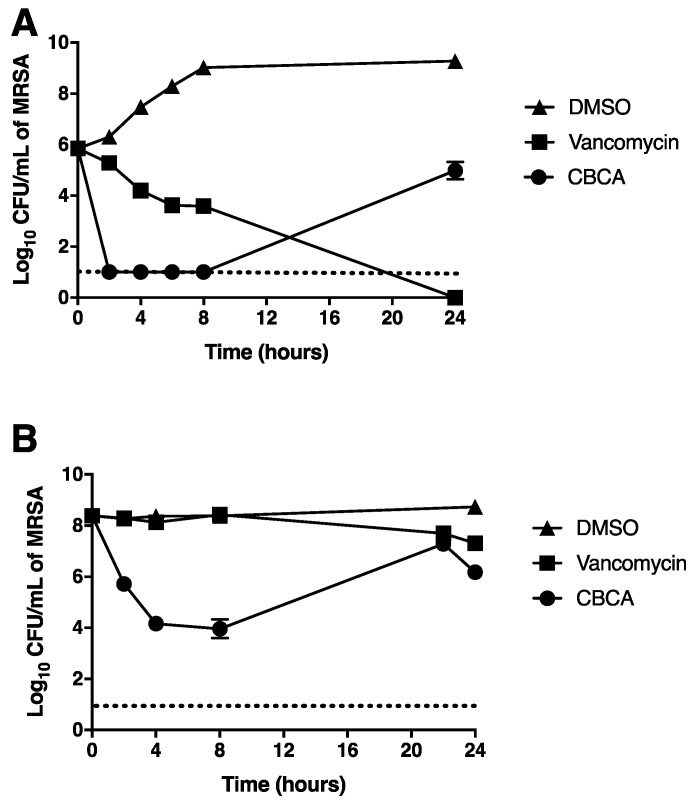
Time-dependent killing of methicillin-resistant *Staphylococcus aureus* (MRSA). MRSA was grown to OD_600_ 0.001 (**A**) or 0.6 (**B**) and incubated with cannabichromenic acid (CBCA) (40 µM), vancomycin (30 µM), or DMSO (1%). At indicated time points, aliquots of cultures were serially diluted and plated for enumeration of viable bacteria (CFU/mL). Error bars represent the standard error obtained from triplicate samples. Dashed lines indicate the level of detection.

**Figure 2 antibiotics-09-00523-f002:**
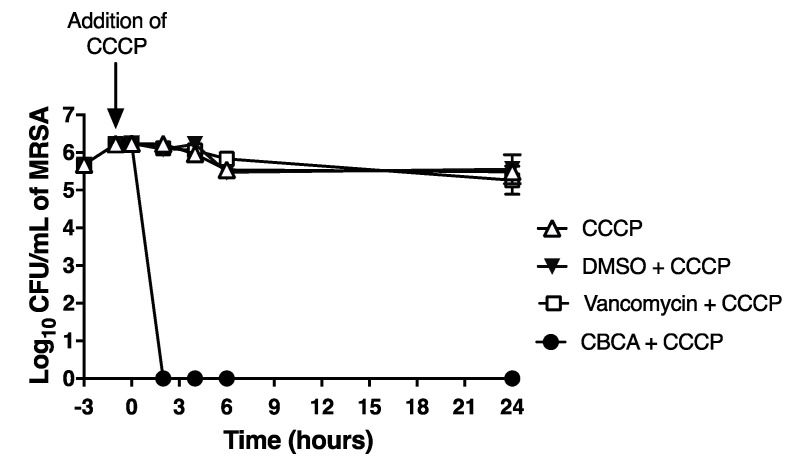
Time-kill assay of carbonyl cyanide *m*-chlorophenylhydrazone (CCCP)-treated MRSA. MRSA were grown to OD_600_ 0.001 and pre-treated with the proton-ionophore CCCP (10 μM) for 1 h to arrest growth. Bacteria were then treated with CBCA (40 μM) or vancomycin (30 μM) for 24 h. At indicated time points aliquots of cultures were retrieved and subjected to serial dilution and plating for enumeration of viable bacteria (CFU/mL). Error bars represent the standard error obtained from triplicate samples. Data representative of two independent experiments.

**Figure 3 antibiotics-09-00523-f003:**
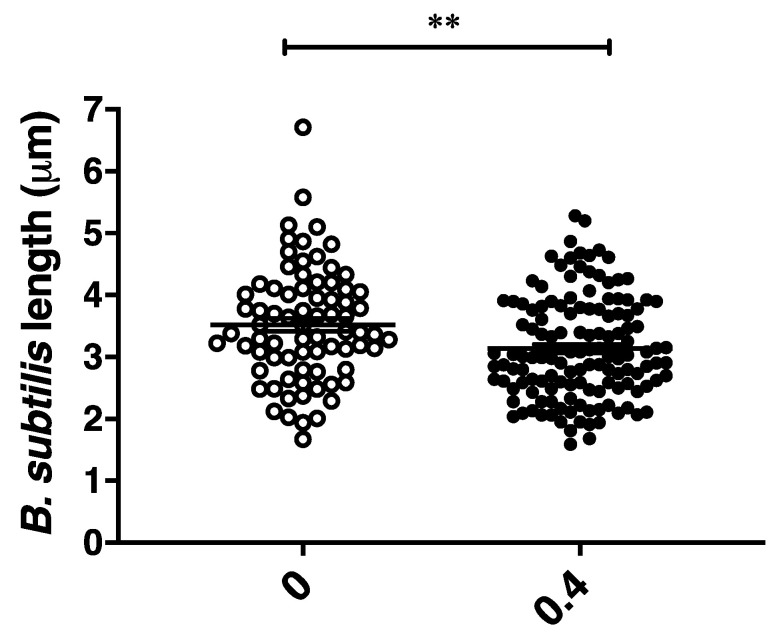
Reduction in *B. subtilis* length following treatment. *B. subtilis* cells in exponential phase (OD_600_ 0.1–0.3) were treated with CBCA at 0.4 × MIC. At 30 min, bacterial cell suspensions were deposited on agarose pads for live phase-contrast microscopy with the Zeiss AxioPlan 2 Upright Light Microscope. Image analysis was performed with AxioVisio (Zeiss). Each symbol represents an individual bacterial cell. Statistical significance determined by one-way ANOVA (** *p* < 0.01).

**Figure 4 antibiotics-09-00523-f004:**
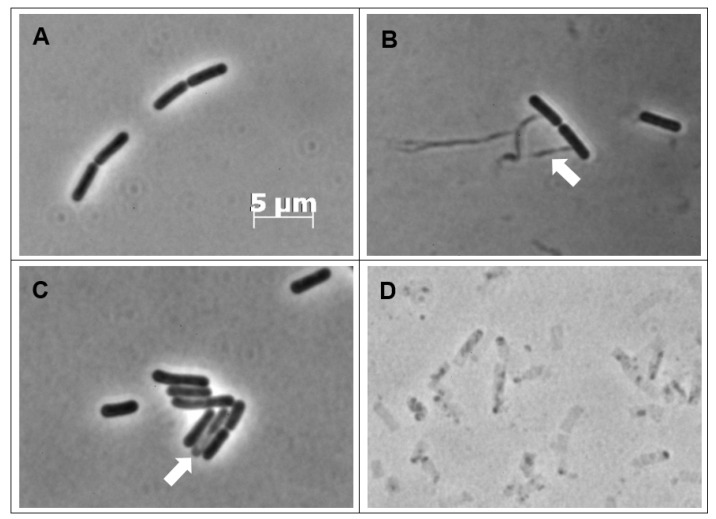
Phase-contrast images of CBCA-treated *B. subtilis* cells. *B. subtilis* cells in exponential phase (OD_600_ 0.1–0.3) were not treated (**A**) or treated with CBCA at 0.25 × MIC (**B**) 0.4 × MIC (**C**) or 1 × MIC (**D**). At 30 min, aliquots were obtained and washed with PBS to remove remaining compound. Bacterial cell suspensions were deposited on agarose pads for live phase-contrast microscopy with the Zeiss AxioPlan 2 Upright Light Microscope and 100 × objective. Image analysis was performed with AxioVision (Zeiss). White arrows indicating insoluble intracellular material leaking from bacteria (**B**) or cells in the process of lysis (**C**).

**Figure 5 antibiotics-09-00523-f005:**
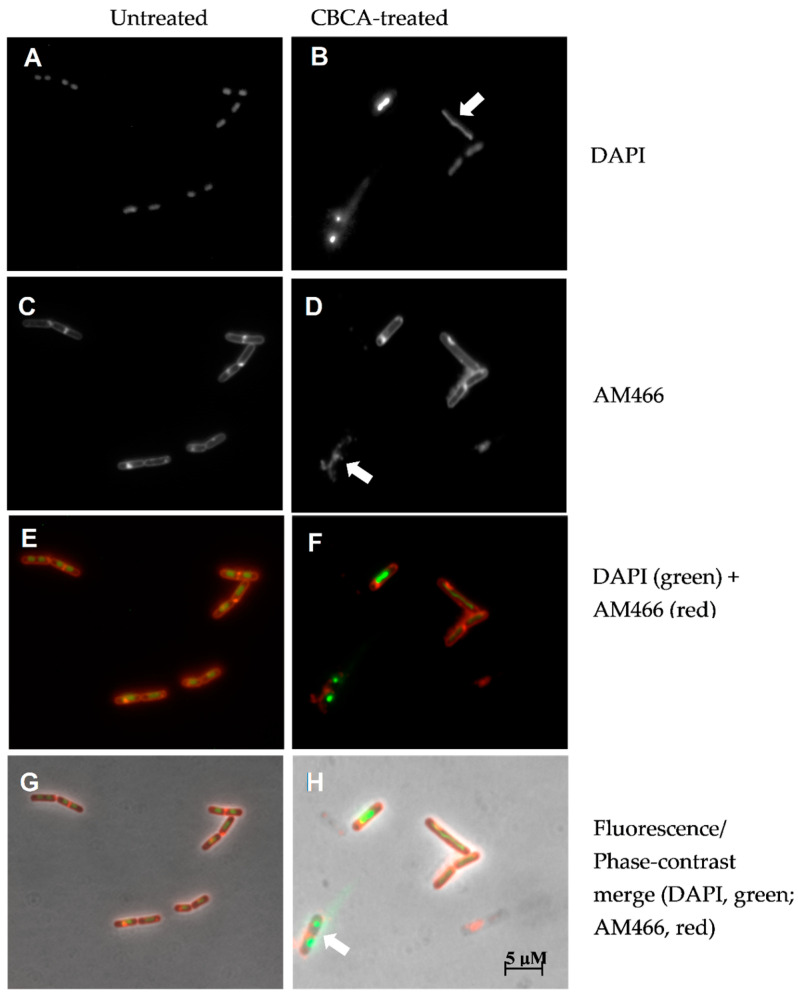
Impact of CBCA treatment on *B. subtilis* cell wall integrity. *B.*
*subtilis* cells in exponential phase (OD_600_ 0.1–0.3) were not treated (**A**,**C**,**E**,**G**) or treated with CBCA at 0.4 × MIC (**B**,**D**,**F**,**H**) as well as the lipid-staining fluorescent molecule AM466 (**C**–**F**). After 30 min, DAPI (200 ng/mL) was added, cell suspensions deposited on agarose pads for fluorescence or phase contrast microscopy. White arrows indicate long-spindle shaped nucleoids and smearing of nuclear material (row 1); abrogation of the lipid membrane (row 2); maintenance of rod-shape of *Bacillus* simultaneous to degradation of nucleoids and lipid membrane (row 4).

**Table 1 antibiotics-09-00523-t001:** Antibacterial activity of phytocannabinoids and related compounds against clinically relevant gram-positive bacteria.

	Antibacterial Activity (MIC; μM) *^a^*			Cytotoxicity (MTC; µM) *^b^*	
Compound	*S. aureus* (MRSA)	*S. aureus* (MSSA)	*E. faecalis* (VRE)	A549	HepG2
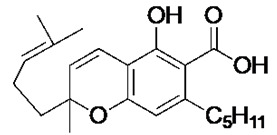 (±)-CBCA	3.9	7.8	7.8	250	125
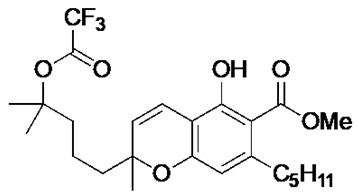 (±)-CBCTFA	>250	>250	>250	*n.d.*	*n.d.*
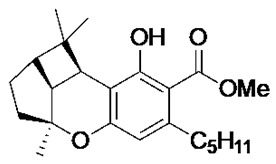 (±)-CBLM	>250	>250	>250	*n.d.*	*n.d.*
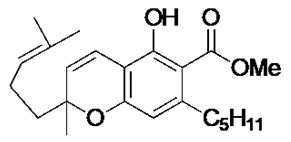 (±)-CBCM	>250	>250	>250	*n.d.*	*n.d.*
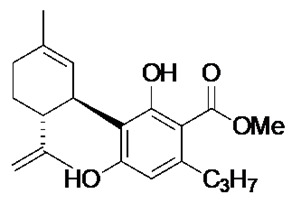 CBDVM	15.6	>250	>250	125	250

*^a^* MIC defined as the lowest concentration of compound required to inhibit total bacterial growth. Assays performed in triplicates. *^b^* One × 10^5^ cells were treated with compounds (0.5 μM–250 μM) for 4 days and MTC (concentration of compound at which cell growth is inhibited) determined. Assay performed in triplicates.

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
