# Peer review of "Rapid Antibacterial Activity of Cannabichromenic Acid against Methicillin-Resistant Staphylococcus aureus"

_antibiotics, 2020, doi:10.3390/antibiotics9080523_

Round 1

Reviewer 1 Report

This is an interesting m/s which reports for the first time the selective activity of a cannabinoid against MRSA; there is extensive data provided by the authors supporting this assertion, and of perhaps greatest interest is that there appears to be a highly novel mode of action, that is, against the bacterial lipid membrane. 

This is potentially a very interesting discovery, but the most potent compound, CBCA, has limited potential for SAR development by medicinal chemists. 

Since this compound is racemic, do the authors know if one enantiomer possesses major bioactivity.  Are the authors able to comment on either of these?

This is an important paper which should be published.

Reviewer 2 Report

The topic of the present  manuscript is the search for novel antimicrobials effective against methicillin-resistant  Staphylococcus aureus (MRSA). To this aim the authors synthesized and tested five phytocannabinoid-based compounds. Among them, one (i.e. cannabichromenic acid – CBCA) proved active invitro against methicillin-sensitive and resistant  S. aureus  and against vancomycin-resistant Enterococcus faecalis (VREF). The topic is highly relevant in the light of the urgent need for novel antibiotics acting with a novel mechanism on as yet less exploited microbial targets. The study is well designed, based on sound methodology and overall well written. However, prior to publication it needs a minor revision as there are some mistakes to be corrected and some missing information to be added. Hence, my opinion is that  it is acceptable for publication after minor revision.

Missing information

  • Bacterial strains: what type of bacterial  strains were  used? Were they ATCC strains? Were they clinical isolates? If yes, provided by whom? How was the antibiotic-resistance determined? In my  opinion, these informations should be provided in Materials and methods.
  • MIC90: The definition of MIC90 provided by the authors is improper as, according to international CLSI and EUCAST guidelines, MIC90 and MIC50 are the minimum concentrations able to inhibit, respectively, 90 and 50% of strains (usually clinical isolates) of one microbial species. The authors should  at least cite a reference in Materials and methods for their alternative definition of MIC90.
  • MIC/MBC: usually when a novel compound is tested for antimicrobial activity, after determination of MIC, the MBC value is next determined. This allows to distinguish between bacteriostatic and bactericidal activity. Since the authors performed  time-kill experiments by using a relatively high concentration of CBCA, does this concentration correspond to MBC? If not, how much higher it is, with respect to MBC? In other words, how much the values of MIC and MBC of CBCA differ?
  • Medium: it is not clear what medium was used in the bacterial assays, LB broth, or TDW (what it is?), or PBS? Please, provide this  information in Materials and methods.
  • Concerning the resazurin assay for cytotoxicity, the cells were incubated with CBCA for 4 days. Considering the CBCA chemical structure, which has some degree of amphipathicity, one could hypotesize a certain ability of this compound to interact with the cell membrane, regardless whether prokaryiotic or eukaryiotic. Hence, it would make sense to look at a shorter incubation time. Have the authors any data concerning acute cytotoxicity (i.e. against mammalian cells) effects after a short incubation time, e.g. 30-60 min?

Minor points/mistakes

Table 1, footnote: delete “highest” and replace with “lowest”; “one x 105 cells” (Table 1, page 3) or “one x 106 cells” (Methods, page 10), which cell concentration is correct?

Figure 1: provide definition of “MK9” or replace “MK9” with “CBCA”.

Figure 2: same comment as for Figure 1.

Figure 5: provide a size bar.

Methods, page 9: in the description of mammalian cell growth conditions, the human embryonic kidney  cells (HEK293T) are cited, but they were not used in the study. Please, remove them from the methods.
